# A Review of Adrenomedullin in Pediatric Patients: A Useful Biomarker

**DOI:** 10.3390/children9081181

**Published:** 2022-08-06

**Authors:** Anna Solé-Ribalta, Sara Bobillo-Pérez, Iolanda Jordan-García

**Affiliations:** Pediatric Intensive Care Unit Service, Hospital Sant Joan de Déu, 08950 Esplugues de Llobregat, Barcelona, Spain

**Keywords:** adrenomedullin, pediatrics, children, biomarker

## Abstract

Adrenomedullin has several properties. It acts as a potent vasodilator, has natriuretic effects, and reduces endothelial permeability. It also plays a role in initiating the early hyperdynamic phase of sepsis. Since its discovery, many articles have been published studying the uses and benefits of this biomarker. The aim of this review is to determine the usefulness of adrenomedullin in pediatric patients. Relevant studies covering adrenomedullin in pediatrics (<18 years) and published up until August 2021 were identified through a search of MEDLINE, PubMed, Embase, Web of Science, Scopus, and Cochrane. Seventy studies were included in the present review, most of them with a low level of evidence (IV to VI). Research on adrenomedullin has primarily been related to infection and the cardiovascular field. The performance of adrenomedullin to quantify infection in children seems satisfactory, especially in sepsis. In congenital heart disease, this biomarker seems to be a useful indicator before, during, and after cardiopulmonary bypass. Adrenomedullin seems to be useful in the pediatric population for a large variety of pathologies, especially regarding infection and cardiovascular conditions. However, it should be used in combination with other biomarkers and clinical or analytical variables, rather than as a single tool.

## 1. Introduction

Adrenomedullin (ADM) is a new, readily measurable circulating biomarker that was first detected in pheochromocytoma cells [1]. The gene locus, part of the calcitonin gene family, is located on chromosome 11. It is a peptide composed of 52 amino acids. After its release, ADM is rapidly eliminated from the serum (half-life of just 22 min); thus, serum levels are difficult to measure [2]. Its precursor molecules are pre-pro-adrenomedullin, which consists of 185 amino acids, and pro-adrenomedullin, which contains 164 amino acids and remains stable for longer periods [3]. In 2004, Struck et al. [4] discovered a more stable mid-regional portion of 52 amino acids, called mid-regional pro-adrenomedullin (MR-proADM). It is co-synthesised with ADM, directly reflects the levels of degradation of active ADM, and has the advantage of a longer half-life and lack of protein binding, which makes it easier to analyse and more suitable for daily clinical practice [5,6]. Recently, assays have become available to measure MR-proADM.

ADM is mainly released from endothelial tissues such as the adrenal medulla, lungs, kidneys, gastrointestinal organs, smooth muscle, and heart [7]. Its production and secretion are stimulated by inflammatory processes, and it has been shown in recent years that cytokines and other inflammatory agents such as LPS, tumour necrosis factor, interleukin-1, and bacterial endotoxins induce its expression.

ADM has several interesting properties [8,9]. Released from endothelial cells, it acts as a potent vasodilator by increasing the synthesis of nitric oxide. It also has natriuretic effects and reduces endothelial permeability [10]. Clinicians have also observed that it plays a key role in initiating the early hyperdynamic response in sepsis and septic shock [11]. The fact that the expression of ADM is stimulated by endotoxins and bacterial cytokines, and that it is bound by the complement control protein Factor H, leads us to assume that ADM has an antibacterial effect and a role as a downregulator of pro-inflammatory cytokines [12].

Since its discovery, many articles in different fields of medicine have been published studying the potential uses and benefits of this biomarker in adults, children, and neonates. To date, no comprehensive review has been undertaken that compiles all of these studied uses in children. We hypothesize that this biomarker can be incorporated into clinical practice for the diagnosis and severity assessment of multiple pathologies.

The aim of this review is to determine the usefulness of ADM in pediatric patients in different situations.

## 2. Methods

To identify relevant studies featuring patients less than 18 years of age, a search was performed using MEDLINE (PubMed), Embase, Web of Science, Scopus, and Cochrane for articles published up until August 2021. This was performed with the search words: (adrenomedullin (title/abstract)) AND children (title/abstract) OR pediatric (title/abstract) OR infant (title/abstract). Reviews, opinion papers, genetics studies, and articles focusing on neonates were excluded. The full texts of the remaining studies, full-text articles assessed for eligibility, were screened by two independent reviewers. Disagreements were resolved through consensus or recourse to a third reviewer. STROBE checklists were used to measure the strength of the articles. The content of this protocol follows Preferred Reporting Items for Systematic reviews and Meta-analysis (PRISMA) recommendations (Please see attached the Appendix A).

Data from reports were extracted by one reviewer. The primary measures considered were the AUC for predicting the outcome, along with the optimal cut-off and the main diagnostic validity parameters (sensitivity, specificity, and positive/negative predictive values). When this information was missing, differences between the means were taken into account. Risk of bias was assessed using the Quality Assessment of Diagnostic Accuracy Studies (QUADAS-2) tool.

Since the study was designed as a systematic review of the literature, permission from an ethical committee was not deemed to be necessary.

## 3. Results

Articles published up until August 2021 involving patients under 18 years old and ADM were reviewed. A total of 126 articles were found. A total of 70 studies were ultimately selected (Table 1, Table 2, Table 3, Table 4, Table 5, Table 6 and Table 7). Most of these studies had a low level of evidence (IV to VI), being case–control or cohort studies. The rest of the studies were rejected for the following reasons: eleven were reviews, seven were articles on adults, two were experimental studies, four were articles on genetics, two were meeting communications, two were opinion papers, one was a preliminary article, ten were articles on neonates, three were articles in a language other than Spanish or English, and fourteen were articles that that did not have a sufficiently high-quality methodology (Figure 1). The summarized QUADAS-2 assessment of the studies is shown in Figure 2. Bias may be present especially regarding patient selection (more than 50% of the studies) and interpretation of the index test (more than 60% of the studies).

### 3.1. Adrenomedullin and Infection

#### 3.1.1. Critically Ill Children and Septic Patients

The available tools to determine the prognosis of critically ill children have mostly been scales developed to estimate the mortality risk based on clinical signs and routine analyses. These include the Pediatric RIsk of Mortality III score (PRISM III) [82], Pediatric Index of Mortality 2 (PIM 2), Pediatric Multiple Organ Dysfunction Score (PMODS) [83], Sepsis-Related Organ Failure Assessment (SOFA) [84], and PEdiatric Logistic Organ Dysfunction-2 (PELOD-2) [85]. These scores are useful to predict the evolution of a wide group of patients within the first 24 h, but their utility in individual patients, especially at the time of admission, seems to be limited. ADM has been recently proposed as a useful biomarker for evaluating disease severity and risk of death.

In a heterogeneous cohort of 254 critically ill pediatric patients recruited from two pediatric intensive care units (PICUs), high levels of MR-proADM, carboxy-terminal pro-endothelin-1 (CT-proET-1) and procalcitonin (PCT) were associated with increased risk of mortality scores as well as an increased risk/number of organ failures. The single marker with the greatest area under curve (AUC) was MR-proADM. Combining MR-proADM with each of the other markers slightly improved diagnostic efficiency. MR-pro-ADM demonstrated an excellent accuracy when detecting patients with higher risk of mortality scores and more than one organ failure, with an AUC of 0.866 (95% CI 0.810–0.821) and 0.922 (95% CI 0.887–0.957), respectively (Table 1). Regarding the sub-cohort of septic patients (n = 40), the AUCs were still excellent (0.869 and 0.901), with similar cut-off points: 0.80 nmol/L (Se 88%, Sp 70%, PPV 23%, NPV 95%) for the prediction of high risk mortality scores and 0.81 nmol/L (Se 90%, Sp 74%, PPV 50%, NPV 96%) for the prediction of more than one organ failure [13].

In 2014, Jordan et al. investigated the prognostic usefulness of MR-proADM in 95 pediatric patients with sepsis admitted to the PICU, finding that MR-proADM levels were significantly higher in those patients requiring mechanical ventilation and vasoactive drugs, although no differences were seen in patients with renal failure. A positive correlation between PRISM III and MR-proADM levels at admission was found (r = 0.447, *p* < 0.001). MR-proADM was also higher in cases of in-hospital mortality (the sample had a mortality of 10.5%), with an AUC of 0.77, slightly lower than the AUC of PRISM III (0.78), and an optimal cut-off point of 2.2 nmol/L. It showed better positive predictive values than PCT and C-reactive protein (CRP) (31% vs. 21.6% and 15.8%, respectively), with similar negative predictive values. In the multivariate study, PRISM III and MR-proADM were identified as independent predictors of in-hospital mortality; thus, their combined use was proposed [14].

Later on, the same group evaluated the diagnostic, prognostic, and stratification potential of a single MR-proADM measurement at the onset of febrile syndrome (Solé-Ribalta et al., 2020). Mean MR-proADM values showed an increasing trend in the transition between non-infected patients, patients with a localized infection, and septic patients, but the difference between non-septic groups and septic groups was not significant. MR-proADM levels increased significantly in patients with severe sepsis (*p* = 0.048) and septic shock (*p* = 0.004) compared with sepsis, with an AUC for the prediction of severe sepsis of 0.729 (*p* = 0.013) and an optimum cut-off of 1.37 nmol/L (Se 79.4%, Sp 50%). Its correlation with mortality risk scores was confirmed. MR-proADM levels were also significantly increased in patients requiring vasoactive drugs; however, unlike in the previous study, they were associated with renal dysfunction but not with the need for mechanical ventilation. It was concluded that PCT appears to be superior to MR-proADM in the diagnosis of sepsis, and that the determination of plasma MR-proADM levels in the early stages of sepsis could be a useful tool for its stratification and predicting morbidity [16].

The results of this last study contrast with the results of Lan J. et al., who also explored the role of MR-proADM in the early diagnosis of childhood sepsis using a sample extracted during the first hour of admission. The authors saw that the diagnostic ability of MR-proADM in children with sepsis was better than that of PCT (AUC 0.869 and 0.757, respectively). Its combined detection effect was better than either single test. MR-proADM and PCT levels also gradually increased in relation to the severity of sepsis and in relation to the systemic inflammatory response syndrome (SIRS) group and control group [15].

#### 3.1.2. Acute Appendicitis

Acute abdominal pain in children is a common reason for visits to the emergency department. Usually, it is a self-limiting process, but in a small percentage of cases, it will require surgery (i.e., acute appendicitis, AA). Diagnosing appendicitis in children is problematic, because most cases show signs and symptoms that mimic other, self-limiting causes of abdominal pain. A delay in the diagnosis and treatment of a case of appendicitis leads to an increase in appendix rupture rates, post-surgical morbidity, mortality, and length of hospital stay, but minimizing the number of negative appendectomies is also important. A large number of inflammatory markers have been used for the early diagnosis of appendicitis. None, in any combination, have shown a satisfactory predictive value for the early diagnosis of appendicitis in the pediatric population. For this reason, some authors have turned their attention to ADM.

According to articles published on this topic, MR-proADM concentrations are higher in children with acute appendicitis than in children with non-specific abdominal pain [17,18]. Greater differences in the levels of this biomarker were seen in complicated cases than in uncomplicated cases of appendicitis, in the first study being not significant (*p* = 0.159), probably due to the smaller number of patients, and being significant in the other (*p* = 0.005). The performance of MR-proADM alone, although statistically significant, was not optimal, with an AUC of 0.75 in the first study and 0.66 in the second, similar to that of CRP (0.72 and 0.70, respectively) and lower than that for leukocyte count (0.88 and 0.84), neutrophil count (0.86 and 0.84), and PAS score (0.87 and 0.76). An MR-proADM level of <0.35 nmol/L, in combination with a low CRP level, seemed useful for identifying children with a low risk of AA, with a sensitivity of >95% and a negative predictive value of >90%. Thus, it seems that ADM alone is not enough for the early diagnosis of AA, but in combination with CRP, it can help to select children with a low risk of AA, and it may contribute to assessing the severity of AA.

#### 3.1.3. Pneumonia

Community-acquired pneumonia (CAP) is a major cause of morbidity and mortality worldwide, especially among children under 5. Basing their opinion only on the assessment of clinical signs and symptoms, clinicians may underestimate or overestimate the severity of the patient’s illness. This can cause unnecessary hospitalization of a patient or trying to treat a patient in an outpatient setting when they require hospitalization, which may cause complications. Due to this, there is a growing interest in searching for biomarkers to improve the risk/severity stratification, alone or in combination with clinical prognostic scores.

Several studies investigating the relationship between pro-ADM levels and CAP patients’ prognosis have been published. Patients presenting complications had significantly higher levels of pro-ADM [19], especially as regards pleural effusion, bacterial pneumonia, empyema, and the need for interventional procedures. Other biomarkers studied, such as copeptin (CoPEP) or interleukin-β1 (IL-β1), showed no added value [20] or inferior results compared to ADM [22]. In this last study, the combination of a Clinical Respiratory Score over 6 points and pro-ADM values over 1.75 nmol/L showed the most significant results (OR: 15.38, 95% CI 1.35–166.66, *p* = 0.027). Despite the homogeneity of the studies, diverse cut-off points have been described: 0.16 nmol/L and 1.75 nmol/L for CAP presenting with complications.

The diagnostic value of ADM in pediatric CAP has only been studied by Esposito et al., who found that MR-proADM blood levels are unable to differentiate bacterial from viral diseases. In this same study, the discriminatory power of MR-proADM to identify complications was found to be insufficient, and no advantages were seen with the use of soluble triggering receptor expressed on myeloid cells-1 (sTREM-1), mid-regional pro-atrial natriuretic peptide (MR-proANP), and MR-proADM, while PCT remained the most useful biomarker for both cases [21].

#### 3.1.4. Pyelonephritis

Urinary tract infection (UTI) is one of the most common infectious pathologies in infancy. Although most cases have an excellent prognosis, when there is renal parenchymal involvement, this leads to an increased associated morbidity and patients may develop proteinuria, high blood pressure, and chronic renal failure in the future. Renal cortical scintigraphy with 99mTc dimercaptosuccinic acid (DMSA scan) is the gold standard for the diagnosis of acute pyelonephritis (APN) and renal scarring (RS). However, it is an invasive test that exposes the patient to radiation and it is not available in all centers. Conversely, there are biomarkers of parenchymal involvement such as CRP or PCT, but none of them are sensitive and specific enough to replace the DMSA scan. For this reason, new molecules that can aid in an early detection of those patients at risk of kidney damage continue to be searched for.

Urine ADM has been studied as a possible solution [23,24,25]. Urine ADM had been observed to be significantly higher in patients with UTI than in healthy controls, and it seems that successful antibiotic treatment of a UTI normalizes ADM levels. In addition, there is a tendency for ADM values to be higher in the urine of patients with ANP than children with acute cystitis. There are conflicting results in the literature about the correlation between white blood cell counts (WBC), CRP, erythrocyte sedimentation rate (ESR), and urine ADM. Sharifian M. et al. proposed a cut-off value of urine ADM for the diagnosis of APN of 100 pg/dL (Se 67.7%, Sp 70%, PPV 70%, NPV 67.7%).

Regarding plasma levels, in one study proADM levels had good predictive value for the diagnosis of APN among patients with UTI (AUC = 0.830, cut-off value for proADM of 63.86 pg/mL [26]. A later study by Peñalver R. et al. could not find a relationship between APN and plasma ADM; however, they did find that plasma MR-proADM had excellent prognostic capacity to predict RS development in patients with APN.

#### 3.1.5. Febrile Neutropenia

Infection is a leading cause of death in children with malignancies. Susceptibility to infection is increased in such patients because they may have to receive intensive chemotherapy, and consequently, they often develop immunosuppression and neutropoenia. The blood culture test, which is considered the gold standard in diagnosing sepsis, is time consuming and associated with a high risk of false negatives (and the blood culture is positive in only up to 20–30% of cases). WBC cannot be applied. The biomarkers that can help uncover the etiological agent can be confounding because of the harmful effects of chemo or radiation therapy. Therefore, a risk score system is commonly used to evaluate and treat patients. However, there is still a need for new, specific biomarkers.

Demirkaya et al. [28] measured serum ADM levels and could not demonstrate a correlation between these levels and the severity of febrile neutropenia. They attributed this phenomenon to its rapid clearance from the serum. Therefore, instead of ADM, they proposed measuring pro-ADM with its longer clearance time. In a study conducted by Fawsi M.M. et al. [31], with severely neutropenic young patients with hematological malignancies, MR-proADM levels were significantly different between the bacteremia/sepsis and pyrexia of unknown origin groups. The area under the receiver operating characteristic curve of MR-proADM had an excellent discriminatory power (AUC = 0.939). The cut-off value defined was 2.4 nmol/L (Se 91.6%, Sp 85.1%, PPV 83.3%, and NPV 92.4%). This study adds to what other recent publications are saying: MR-proADM is a promising early biomarker for sepsis in severely neutropenic young patients with hematological malignancies. Regarding risk prediction, ADM levels at admission seem to be useful in identifying high-risk patients with solid tumors [29], and they have been correlated with length of stay (LOS) in hospital [30].

### 3.2. Cardiovascular Disease

#### 3.2.1. Congenital Heart Disease

Studies have demonstrated that ADM has various physiological effects on the cardiovascular system, including vasodilatation, diuresis, natriuresis, inhibition of aldosterone secretion, and increased cardiac output. These findings suggest that plasma ADM may participate in the regulation of circulatory homeostasis and in the pathophysiology of cardiovascular disease. Regarding the behavior of this biomarker in congenital heart disease, it seems that ADM levels may play beneficial roles in reducing increased pulmonary arterial resistance or alleviating hypoxemia in these patients, since its levels are higher in patients with congenital heart disease (CHD) and pulmonary hypertension (PH), and it is positively correlated with pulmonary arterial resistance (Rp). ADM levels are also higher in cyanotic congenital heart disease and are negatively correlated with systemic arterial oxygen saturation (SA_sat_) and mixed venous oxygen saturation. In addition, its levels were lower in the pulmonary vein than in the pulmonary artery, suggesting that it is extracted in pulmonary circulation [36,37,38].

#### 3.2.2. Cardiac Surgery

All organs are harmed during cardiopulmonary bypass, and the pulmonary endothelium is one of the most severely damaged, causing high pulmonary vascular resistance, among other consequences. ADM, an intrinsic vasodilator thought to act as a circulating hormone that regulates vascular tone, has been analysed in several studies involving children undergoing open-heart surgery. Plasma ADM levels significantly increase after cardiopulmonary bypass (CPB), but patients with high pulmonary arterial flow showed lower levels than other patients, both pre- and post-CPB. One explanation could be previous endothelial damage in these patients, also supported by the correlation between post-CPB ADM levels and pre-CPB mean pulmonary arterial pressure (mPAP) [39].

In addition, the preservation of the myocardial tissue remains a challenge, and the distribution of cardioplegia solution is altered due to partial vasoconstriction. Thus, it was thought by Szekely et al. that the presence of vasodilators such as ADM could improve this distribution. Their results showed that preoperative ADM levels in patients with cardiac troponin I (cTn-I) levels <10 ng/mL after the surgery were significantly greater than those with either moderate or severe injury; thus, their hypothesis was sustained [40].

The CPB procedure in itself implies a complex fluid homeostasis with the involvement of many hormones, such as vasopressin, atrial and brain natriuretic peptides, and aldosterone. ADM kinetics during CPB has also been studied. ADM levels increased gradually, peaking 60 min after CPB, and then decreased 24 h after the operation, in tandem with changes in urine sodium and vasopressin. Besides these two parameters, ADM was also correlated with urine volume, plasma osmolarity, and plasma brain natriuretic peptide (BNP). Changes in plasma ADM levels seem to correspond to fluid and electrolyte balance, and ADM may play a role in fluid homeostasis, in cooperation with other hormones, during CPB [41].

CPB affects the cerebrovascular autoregulation system as well, mainly due to the hypoperfusion–reperfusion sequence, embolization, and/or thermal injury. Florio et al. found that at the end of CPB and surgery, ADM was significantly lower in children in whom brain damage was shown during follow-up for cardiovascular surgery; these children also had increased cerebrovascular resistance. They proposed a cut-off value of 17.4 ng/L for ADM at the end of CBP for predicting brain damage [42].

ADM has been proposed as a biomarker for low cardiac output syndrome (LCOS) after CPB in children, which would be of great interest because this is still a major perioperative complication. ADM levels in patients that developed LCOS were significantly lower than those who did not, reaching a dip at the end of the surgical procedure. One multivariable analysis demonstrated that ADM levels and cooling were independent variables for predicting LCOS, and a cut-off point of 27 pg/L for ADM at the end of the surgery was proposed for this [43]. In another study, cTn-I levels (>14 ng/mL; OR 4.05, 95% CI 1.29–12.64, *p* = 0.016) at 2 h following corrective heart surgery under CPB and MR-proADM levels (>1.5 nmol/L; OR 15.54, 95% CI 4.41–54.71, *p* < 0.001) at 24 h post-operation were independent predictors of LCOS [45]. This, however, differs from two previous papers [41,43] that did not find an initial decrease in ADM levels during and after CPB in children with LCOS. One year later, this same author published that a vasoactive inotropic score (VIS) of >15.5 at 2 h post-CPB, adjusted for age and CPB timepoints, showed a high specificity (92.87%) and negative predictive value (75.59%) for diagnosing LCOS at 48 h post-CPB, although the predictive power for LCOS did not increase when VIS was combined with cTn-I >14 ng/mL at 2 h and MR-proADM >1.5 nmol/L at 24 h post-CPB. This could provide important information for physicians so they can engage in early interventions and optimally manage these patients [46].

Regarding the need for intensive support and complications after CPB, MR-proADM and pro-atrial natriuretic peptide (pro-ANP) were associated with the need for MV, with respective cut-off values of 1.22 nmol/L and 215.3 pmol/L potentially indicating mechanical ventilation (MV) requirement with respective AUCs of 0.721 and 0.746 at admission and 0.738 and 0.753 at 24–36 h. PCT levels greater than 1.9 ng/mL at 24–36 h post-CPB were predictive for invasive bacterial infection (IBI), with an AUC of 0.896 (Se 95.5, Sp 95.5, PPV 79.1, NPV 96.2) [47]. Pro-ANP and MR-proADM levels before cardiac surgery were also higher in those patients with an increased need for ventilatory and inotropic support after the surgical intervention. The AUC for pro-ANP was greater than that of MR-proADM, but without significant differences for both cases; however, in the multivariate analysis, the only biomarker kept as an independent predictor was pro-ANP [48].

#### 3.2.3. Fontan Procedure

Pulmonary factors, especially pulmonary vascular resistance (PVR), are known to be crucial when deciding on the Fontan procedure. Therefore, the cytokines that influence pulmonary circulation also seem to influence hemodynamics after the Fontan procedure. Immediately after the Fontan procedure, both endothelin-1 (ET-1) and ADM are elevated, but plasma levels of ET-1 are maintained and then significantly increase 6 and 24 h after CPB, unlike ADM. Therefore, the imbalance of these two cytokines may trigger vasoconstriction after the Fontan procedure that may compensate for the relatively low cardiac output [49]. ADM has also been studied during follow-up on Fontan procedure patients. Higher concentrations were found in these patients than in controls, and a negative correlation was also observed between ADM levels and cardiac output [50]. MR-proADM levels may help identify patients at risk for a failed Fontan, and a cut-off of 0.520 nmol/L has been proposed for predicting Fontan failure [51].

#### 3.2.4. Heart Failure and Dilated Cardiomyopathy

Measuring blood biomarkers can facilitate heart failure (HF) management, as they provide objective information on disease severity, prognosis, and treatment response. Although plasma levels of ADM were increased in pediatric and adult patients with CHF irrespective of the cause [52], a larger study focusing on the pediatric population found an unsatisfactory diagnostic power for MR-proADM regarding general patients with HF and the subset of patients with dilated cardiomyopathy (DCM) [53]; it was, however, negatively correlated with poor left ventricle (LV) function. In this study, MR-proANP was found to be the best biomarker discriminating HF in children and adolescents with CHD and CMP, even if well compensated, and its diagnostic performance was comparable to that of NT-proBNP. Conversely, another study including only a small sample of children with DCM observed significantly lower levels of ADM in DCM patients than controls. They hypothesized that this finding could be due to events such as the necrosis and apoptosis of cardiomyocytes in advanced stages of heart failure, but this hypothesis must be examined in additional studies. Despite their different results, they agree on the negative correlation between ADM levels and ejection fraction (EF) and fractional shortening (FS) [54].

#### 3.2.5. Postural Orthostatic Tachycardia Syndrome

Children with postural orthostatic tachycardia syndrome (POTS) experience symptoms of orthostatic intolerance in association with excessive tachycardia. Midodrine hydrochloride, a vasoconstrictor, has been reported to improve symptoms, although the response rate is around 70%. A predictor for its response has not been described, but it is presumed that children with abnormal vascular resistance might have a positive response to the drug. In 2012, Zhang et al. studied the predictive value of MR-proADM for assessing the therapeutic efficacy of midodrine hydrochloride. They found that those responding positively to midodrine hydrochloride had higher plasma levels of MR-proADM than non-responders, with an excellent discriminatory power (AUC of 0.879), and they proposed a cut-off of 61.5 pg/mL to predict the efficacy of the drug [55]. The same group reported that patients with levels above the defined cut-off had a significantly lower symptom score at the 60-month follow-up and a significantly higher symptom-free survival at the 72-month follow-up [56].

### 3.3. Pulmonary Hypertension

ADM exerts its vasodilatory activity on a number of vascular beds, including the lungs, and stimulates cyclic adenosine monophosphate (cAMP) production in vascular smooth muscle cells, suggesting that it may play an important role in modulating PH. An increased pulmonary vascular resistance has been documented during and after CPB and this is a major concern in children with CHD.

ADM is significantly elevated in patients with PH (primary and related to congenital heart disease) when compared to those without PH, and a significantly increased uptake of plasma ADM in pulmonary circulation has been demonstrated in these patients [58]. Additionally, as the severity of PH increases, the plasma levels of ADM also increase in tandem (the same as ET-1 and contrary to nitric oxide (NO) levels), and a correlation between ADM levels and PAP was observed [61,62]. These findings suggest the involvement of ADM in the pathophysiology of PH. ADM may be involved in the defense mechanism against further increases in pulmonary arterial pressure and may be a reliable method to monitor changes in pulmonary pressure and worsening PH.

With respect to the presurgical stage, Vijay et al. demonstrated that preoperative plasma ADM levels are lower in patients who are at a higher risk for the development of postoperative PH (patients with high pulmonary flow), suggesting that the impaired ability to synthesize ADM in the pulmonary circulation may contribute to the risk of developing PH in pediatric patients undergoing CPB [59].

As for long-term follow-up, ADM may be valuable for evaluating pulmonary hemodynamics after long-term treatment with PGI2 in primary PH [60].

### 3.4. Nephro-Urological Disease

ADM is a potent vasodilator with proven antimitogenic and antiproliferative effects in renal mesangial cells, as well as diuretic and natriuretic actions. Plasma and urine ADM levels are known to deviate from normal levels in many renal diseases.

Balat et al. studied plasma and urine levels of ADM in minimal change nephrotic syndrome (MCNS) [64] and found that ADM levels were significantly lowered in plasma and were elevated in urine in these patients during relapse. This is possibly due to ADM loss through the urine secondary to severe proteinuria and because the kidney may be one of the major sites for ADM synthesis. In addition, NO and ADM, with their vasodilatory effects, were considered of interest in children with detrusor instability (DI) [66]. It was found that a decreased NO production most likely has a role in the pathophysiology of DI, although increased ADM appears to be compensatory, and the functional significance of ADM and NO in bladder smooth muscle remains to be determined by further, more in-depth studies. The same author studied the role of NO and ADM in Bartter syndrome [63], in which patients are known to have a lower vascular reactivity. It was found that urinary nitrite and ADM excretion was lower in Bartter syndrome patients, suggesting a potential role of these two molecules in the reduced vascular response seen in the disease. Finally, Balat and his group investigated the role of ADM and NO in primary nocturnal enuresis [65], finding that bed-wetting children had plasma levels that were significantly lower than the controls, suggesting a deficient synthesis in these children.

In 2005, Kalman et al. studied ADM in children with urinary tract infections with renal parenchymal scars (RPS) and vesicoureteral reflux (VUR) [67]. Plasma and urine ADM levels were higher, although not significantly, in the control group. The authors justify this finding with the fact that prophylactic antibiotics may lower ADM secretion. However, the low urinary ADM levels of patients with renal parenchymal scars and preserved renal function might be predictive of renal injury, and the lower ADM levels in patients with a higher degree of VUR may indicate that ADM has a role in the ureteral antireflux mechanism. Thus, they conclude that ADM can be a prognostic factor in the long-term follow-up of cases with these diseases.

### 3.5. Endocrine Pathology

Obese children are at an increased risk for abnormal cardiac structure and function and other metabolic risks. Studies on adipokines and other biomarkers of obesity have become important in obesity research, and ADM was also defined as a new member of the adipokine family. ADM secreted by adipocytes, through its vasodilator and antioxidant actions, might be protective against metabolic-syndrome-associated cardiovascular complications.

Plasma MR-proADM levels are significantly higher in obese than in normal-weight adolescents, and its levels are correlated with BMI z-score, fat mass, circulating insulin, HOMA-IR, total cholesterol, and LDL cholesterol, suggesting its important involvement in obese patients [68]. A positive correlation between ADM levels and left ventricle mass index (LVMI) have also been demonstrated (LVMI OR 1.14, 95% Cl 1.08–1.13, *p* = 0.0001). In this same article, a cut-off value of ADM at 52 pg/mL is proposed to differentiate obese children with and without left ventricular hypertrophy, yielding a sensitivity of 94.32% and specificity of 92.45%. Thus, measuring plasma ADM levels in obese children may help to identify those at high risk of developing LV hypertrophy and dysfunction [69].

Due to its vascular effects, including endothelial vasodilatation, antioxidative stress, stimulation of endothelial nitric oxide production, and antiproliferation of vascular smooth muscle cells and adventitial fibroblasts, it has been speculated that ADM could have a role in microvascular complications in patients with type 1 diabetes mellitus. However, the two articles available are contradictory as regards whether its levels would be increased or decreased.

### 3.6. Rheumatic Diseases

The main source of circulating ADM is now thought to be the vasculature. As both vascular endothelial cells and smooth muscle cells prominently express ADM and its receptor, ADM may control vascular functions and may have a role in rheumatic diseases such as Kawasaki disease (KD), Henoch–Schönlein purpura (HSP), acute rheumatic fever (ARF), and Familial Mediterranean Fever (FMF).

The formation of coronary artery aneurysms is a major complication of KD, and early detection and intensive treatment for it is important. ADM levels were markedly elevated before treatment, especially in patients with coronary artery dilation. They substantially decreased after treatment. Thus, it may be a useful biomarker to monitor KD patients during the acute phase and may help to diagnose coronary artery involvement [72].

ADM may play a role in endothelial injury in HSP since there are higher plasma and urine levels of ADM in patients in the acute phase versus in remission and controls. However, whether this perpetuates, or protects against, further vascular injury is not clear [73].

ARF has the characteristics of an autoimmune disease, triggered by cross-reactive antigens shared by group A streptococci and involving a variety of tissues, such as the heart and endothelium. Plasma and urine ADM levels were significantly higher in children with ARF, irrespective of whether they were in the acute or convalescent phases. ADM may play a role in the immunoinflammatory process of ARF. By contrast, these increased levels may also be the result of inflammatory injury in ARF [54].

FMF is characterized by recurrent attacks of polyserositis. As ADM is synthesized in the endothelium and mediates many functions within the immune system, it has been considered an interesting FMF research target. Plasma and urine levels were significantly higher in patients than in controls; thus, ADM may have a role in the immuno-inflammatory process of FMF, although whether it acts to sustain or protect against further inflammatory injury is not clear [74]. Renal amyloidosis, which is the most important FMF complication, also determines its prognosis. Although information is still conflicting, M694V homozygosity is a risk factor for amyloidosis; thus, the relationship between this genotype and ADM has been studied. Although these results indicated higher ADM levels in patients with a homozygous M694V mutation, further studies are needed to relate ADM levels and renal amyloidosis (62). Finally, in order to investigate the subclinical inflammation present between FMF attacks, Polat et al. conducted a clinical trial with FMF patients at different colchicine dosages. No changes in ADM were demonstrated, suggesting the continuation of the clinical inflammatory process in these patients [76].

## 4. Discussion

From this literature review, we can see that research on ADM has focused on infection and cardiovascular diseases. Most of the studies had a low level of evidence (IV to VI), being case–control or cohort studies. Measurements of ADM are quite heterogeneous, in terms of the ADM form (adrenomedullin, pro-adrenomedullin, mid-regional pro-adrenomedullin, and even sub-forms), dosing techniques, and samples chosen for measurement (plasma, serum, urine, tissues, and CSF).

In general, the performance of ADM as a biomarker for pediatric infection seems satisfactory. In pediatric sepsis, it seems to be useful for stratification, prediction of organ dysfunction, and mortality risk prediction, with consistent cut-off points [14,15,16]. These results are similar to those found in the adult population where several studies show that MR-proADM can be helpful in the individual assessment of risk in septic patients admitted to intensive care units (ICU). In 2006, Christ-Crain et al. [86] concluded, after evaluating 53 patients admitted to an ICU for sepsis, that MR-proADM levels at ICU admission are able to predict the patient’s condition with an accuracy similar to that of APACHE II and SAPS II. Suberviola et al. [87] showed that MR-proADM is a better tool for predicting in-hospital death than C-reactive protein or procalcitonin in patients admitted to the ICU. Its capacity to identify invasive bacterial infections in well-appearing febrile infants is poor, but this improves in combination with other biomarkers such as PCT [33]. Regarding pneumonia in the adult population, in the study by Cavalli R. et al. [88] from 2014, MR-proADM levels were correlated with mortality; Pereira J.M. et al. [89], remarked that patients with severe CAP who have decreased levels of MR-proADM after 48 h of antibiotic treatment have a better vital prognosis, as well as the study by Akpinar S. et al. [90], who observed that the determination of MR-proADM alone does not correlate with mortality or disease severity to produce mortality but that the union of SOFA, APACHE II and MR- proADM may be good prognostic markers in CAP patients. In pneumonia, it seems to identify patients with complications, although with differing cut-off points [19,20,22]. In cancer patients with neutropoenia, it does not predict a positive blood culture, but it appears to identify high-risk patients and sepsis [28,29,31]. More research is needed in this regard, and it is potentially necessary to differentiate between patients with solid tumors and patients with hematological malignancies. In patients with acute abdomen, its combined use with CRP is proposed to identify low-risk patients [17,18]. Finally, in UTIs, it seems to identify patients with pyelonephritis and patients with pyelonephritis who will develop renal scarring [26,27].

Regarding the behavior of this biomarker in CHD, it seems that ADM levels may play beneficial roles in reducing increased pulmonary arterial resistance or alleviating hypoxemia [37]. Conflicting results exist in the literature regarding the peri-CPB kinetics of ADM. Measuring pre-CPB levels could be useful to predict myocardial tissue damage [40]. During the surgery, ADM levels seem to correspond to fluid and electrolyte balance and may help to predict brain damage [41]. After the surgery, ADM concentrations were useful to predict LCOS and the need for respiratory and inotropic support [43,45]. Vasoconstriction is common after the Fontan procedure; low ADM levels have been found immediately after this surgery, and this has been understood as being protective when there is low cardiac output. Conversely, high levels of this biomarker in the long term may help to identify patients at risk for a failed Fontan [51]. Several studies have analysed the role of adrenomedullin in heart failure in adults, which is linked to cardiac remodeling under stress [91,92]. The higher levels of adrenomedullin were associated with greater volume overload [93], both intravascular and interstitial. In these patients, linked to this situation of overload, the elevated level of adrenomedullin was related to a higher risk of morbidity and mortality [94,95]. In pediatrics, studies regarding heart failure are scarce; unsatisfactory results of ADM were observed in identifying heart failure and dilated cardiomyopathy [96]. Regarding postural orthostatic tachycardia syndrome, ADM could identify those patients who would respond well to vasoconstrictive therapy [55,56].

In PH, ADM may be involved in the defense mechanism against further increases in pulmonary arterial pressure, since levels are elevated in these patients and they negatively correlate with pulmonary arterial pressure [61]. Nevertheless, one article found lower ADM levels in patients at risk of PH (high pulmonary flow patients), and they justify that the impaired ability to synthesize ADM in the pulmonary circulation may contribute to the risk of developing PH in pediatric patients undergoing CPB [59].

Studies on nephro-urologic and rheumatic diseases are scarce and heterogeneous. ADM levels in childhood have been studied in Bartter syndrome, minimal change nephrotic syndrome, primary nocturnal enuresis, detrusor instability, and VUR [63,64,65,66,67]. This peptide seems to have a role in all these pathologies, and in the case of VUR, it seems to identify patients with parenchymal scarring. ADM may have a role in the immune-inflammatory process of HSP, ARF, and FMF, although how they affect inflammatory injury is not clear [73,75,76]. In KD, this biomarker may be useful to monitor patients during its acute phase and may help to determine coronary artery involvement [72].

Our systematic review has some limitations. Most of the studies included had a low level of evidence (IV to VI), being case–control or cohort studies. QUADAS-2 analysis revealed that bias may be present in more than 50% of the studies, especially regarding patient selection and interpretation of the index test. In many cases, the primary measures considered, AUC for predicting the outcome, optimal cut-off and the main diagnostic validity parameters were missing. This fact, along with the heterogeneity in the forms of adrenomedullin measured, made it unfeasible to present a regulated synthesis of results.

## 5. Conclusions

ADM seems to be useful in the pediatric population for a large variety of pathologies, but it is especially valuable in infection and cardiovascular conditions. Taking into account all of the above, MR-proADM should be used in combination with other biomarkers and clinical or analytical variables rather than as a single tool. The current clinical data are still limited in some pathologies and have a low level of evidence. More studies are needed to confirm the usefulness of measuring this peptide in some fields.

## Figures and Tables

**Figure 1 children-09-01181-f001:**
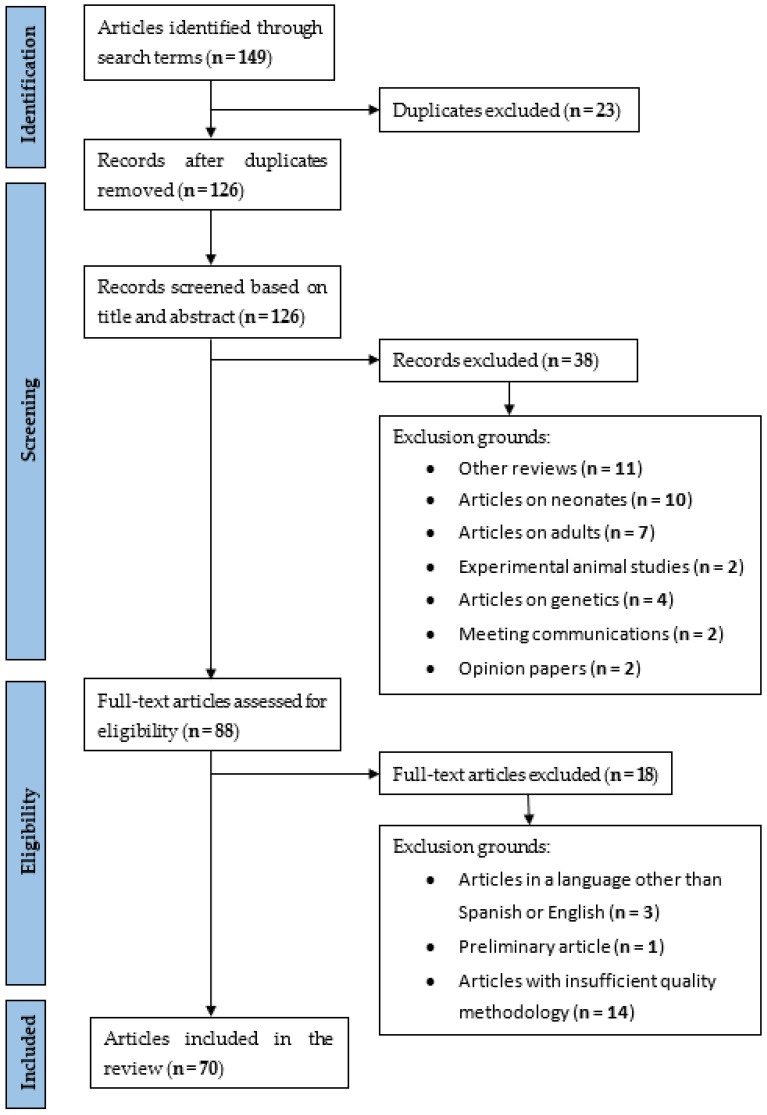
Flow diagram.

**Figure 2 children-09-01181-f002:**
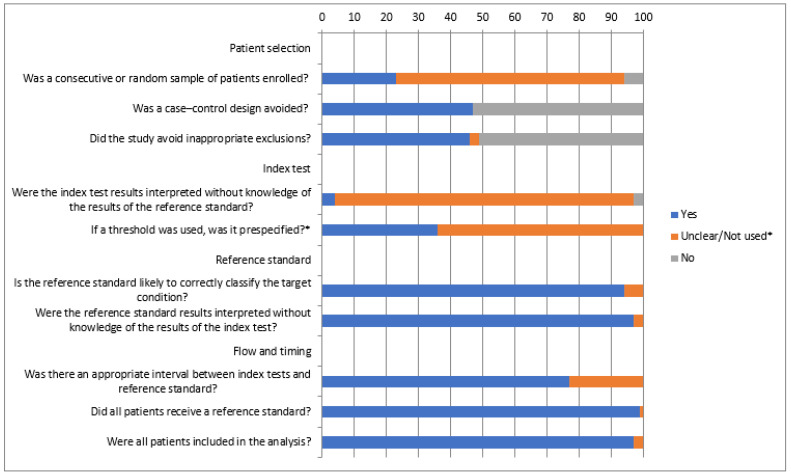
Summary of the quality of the studies included in this review. * means that in the question “if a threshold was used, as it prespecified?”.

**Table 1 children-09-01181-t001:** Adrenomedullin in critically ill children and in paediatric infections.

Study	Type of Study	Setting	Biomarkers Studied	Aim	Gold Standard	AUC	Cut-Off	Se	Sp	PPV/ NPV	Conclusion
**Critically Ill Children**											
Rey C. et al. (2013) [13]	P, O, MC (2PICU)	<18 yr *n* = 254 PICU patients	MR-proADM CT-proET-1 PCT CRP	To identify biomarkers that can be used as predictors of mortality risk.	PRISM III and PIM mortality risk scores, no. of organ failures	0.866 0.922	0.79 nmol/L (scores) 0.77 nmol/L (organ failure)	93 91	76 85	33.7/98.2 69.3/95.6	MR-proADM appears to improve diagnostic accuracy for detecting patients with higher risk of mortality scores and more than one organ failure.
**Sepsis**											
Jordan I. et al. (2014) [14]	P, O, SC	9 d–13 yr *n* = 95 Septic PICU patients	MR-proADM PCT CRP	To determine MR-proADM’s prognostic usefulness.	Organ failure, PRISM III, and in-hospital mortality	0.77	2.2 nmol/L (mortality risk)	72.7	81	31/96.2	MR-pro-ADM levels are good indicators of disease severity and show better reliability than PCT and CRP for predicting in-hospital mortality.
Lan J. et al. (2019) [15]	CC, SC	6–12 yr *n* = 139 (94 septic, 25 SIRS, 20 controls)	PCT MR-proADM	To evaluate the role of MR-proADM and PCT in the early diagnosis of childhood sepsis.	-	0.869	3.46 mmol/L (sepsis diagnosis)	85.11	71.1	-	PCT and MR-proADM concentrations significantly increase as severity of sepsis worsens. The diagnostic effect of MR-proADM in children with sepsis was better than that of PCT. Their combined detection effect was better than either by itself.
Solé-Ribalta A. et al. (2020) [16]	P, O, SC	<16 yr *n* = 75 Febrile patients with suspected sepsis	PCT MR-proADM	To evaluate the diagnostic, prognostic, and stratification potential of MR-proADM at the onset of fever.	Goldstein 2005 diagnostic criteria	0.729	1.37 nmol/L (severe sepsis)	79.4	50	79.4/50	PCT appears to be superior to MR-proADM in diagnosing sepsis; MR-proADM in the early stage of sepsis could be a useful tool for the stratification of sepsis and the prediction of morbidity.
**Acute Appendicitis**											
Míguez et al. (2016) [17]	P, O, SC	3–16 yr *n* = 136 ED patients with suspicion of AA	CRP WBC and neutrophil count MR-proADM PAS score	To evaluate the usefulness of MR-proADM in diagnosing AA in children presenting with acute abdominal pain.	Histological confirmation on surgeon’s report	0.75	0.34 nmol/L (to rule out AA diagnosis)	93	46	45/93	The performance of MR-proADM alone, while statistically significant, is not optimal. MR-proADM levels of <0.35 nmol/L in combination with low CRP seems useful for the identification of children with a low risk of AA.
Oikonomopoulou et al. (2019) [18]	P, O, MC (6 ED)	<18 yr *n* = 285 ED patients with suspicion of AA	MR-proADM CRP WBC and neutrophil count	To investigate the utility of pro-ADM for diagnosing AA.	Histological confirmation on surgeon’s report	0.66	0.35 nmol/L (to rule out AA diagnosis)	92	32	43/88	MR-proADM alone is not enough to diagnose AA early. The combination of low values of MR-proADM and CRP can help select children with a low risk of AA.
**Pneumonia**											
Sardà M. et al. (2012) [19]	P, O, SC	<18 yr *n* = 50 ED patients admitted with CAP	pro-ADM	To determine the levels of pro-ADM in children with CAP and analyse the relationship between these levels and the patients’ prognosis.	Radiological imaging	The median level of pro-ADM was 1.0065 nmol/L (IQR 0.3715–7.2840) The patients presenting complications had higher levels of pro-ADM (2.3190 vs. 1.1758 nmol/L, *p* = 0.013). Specifically, the presence of pleural effusion was associated with higher levels of pro-ADM (2.9440 vs. 1.1373 nmol/L, *p* < 0.001).					Higher levels of pro-ADM at admission were related to a greater likelihood of complications during the hospital stay, especially pleural effusion.
Alcoba G. et al. (2015) [20]	P, O, MC (3 ED)	<16 yr *n* = 88 Presenting at ED with CAP	pro-ADM CoPEP	To assess the diagnostic accuracy of pro-ADM and CoPEP for predicting CAP complications in children.	Culture, chest X-ray	0.72	0.16 nmol/L (complications)	72.7	71.4	26.7/94.8	Proadrenomedullin seems to be a reliable and readily available predictor for CAP complications.
Esposito S. et al. (2016) [21]	P, O, MC (N = 11)	4 mo-14 yr *n* = 433 Children hospitalised with CAP	sTREM-1 MR-proANP MR-proADM	To evaluate the diagnostic accuracy of biomarkers to distinguish bacterial from viral CAP and to identify severe cases of CAP.	Blood and respiratory samples, real-time PCR, chest X-ray, BTS criteria	0.58 0.55	0.32 nmol/L (bacterial infection) 0.39 nmol/L (severe disease)	78.0 51.4	35.7 66.1	59.8/57.0 7.5/34.7	MR-proADM blood levels cannot be used to differentiate bacterial from viral diseases or to identify severe cases.
Korkmaz M.F. et al. (2018) [22]	P, O, SC	3 mo-18 yr *n* = 66 Children hospitalised with CAP	pro-ADM IL-1β	To investigate the value of Pro-ADM and IL-1β for severity assessment and outcome prediction in children with CAP.	Chest imaging, culture, Respiratory Clinical Score	- 1.00	1.75 nmol/L (complications) 4.95 nmol/L (high-severity group)	90 100	66.6 92	- 78.9/92.1	Pro-ADM may offer additional risk/severity stratification in children with CAP and may be helpful in predicting the development of complications (need for PICU admission and intervention procedures).
**Pyelonephritis**											
Dötsch J. et al. (1998) [23]	CC, SC	2 w–8 yr *n* = 22 (11 patients with UTI, 11 controls)	Urine ADM	To investigate whether UTIs are associated with increased urine ADM excretion.	Culture	ADM levels in children with UTIs were significantly higher than in controls (0.6 ± 0.41 vs. 0.15 ± 0.14 ng/μmol creatinine *p* < 0.001) There was a significant correlation between WBC count and ADM in urine (r = 0.78, *p* < 0.001).					Urinary tract infections possibly promote increased ADM release from urinary tissue, although the mechanism is unclear.
Kalman S. et al. (2005) [24]	CC, SC	6–18 mo *n* = 35 (19 patients with APN, 16 controls)	Plasma and urine ADM	To determine plasma and urine ADM levels in children with APN and compare the results with a control group.	Culture, DMSA	The plasma ADM levels were lower in APN patients than in the control group (*p* < 0.001). Urine ADM levels were higher in APN patients than in the control group (*p* < 0.001). The correlations between urine ADM levels and C-reactive protein and white blood cells were statistically significant.					Urine ADM may have a role in APN. The importance of plasma ADM in the pathogenesis of acute pyelonephritis remains to be determined.
Sharifian M. et al. (2013) [25]	CC, SC	1 mo-10 yr *n* = 60 (31 with APN, 30 healthy controls)	Urine ADM	To evaluate the association between APN and urine ADM, as well as the effect of treatment on ADM levels.	DMSA	-	Urine ADM > 100 pg/dL (APN diagnosis)	67.7	70	70/67.7	This marker can be used for confirming the diagnosis and evaluating response to treatment in combination with other biomarkers.
Cetin N. et al. (2020) [26]	CC, SC	2–18 yr *n* = 82 (38 with APN, 24 with UTI, 20 healthy controls)	PSP pro-ADM TREM-1	To investigate the diagnostic values of presepsin, pro-ADM, and TREM-1 levels in children with APN and lower UTI.	Culture, standard biomarkers (ESR, CRP, WBC)	0.83	63.86 pg/mL (APN prediction)	-	-	92.1/87.5	Plasma pro-ADM could be a useful biomarker for the early diagnosis of acute pyelonephritis in children.
Peñalver R. et al. (2021) [27]	P, O, SC	<16 yr *n* = 62 Patients diagnosed with fUTI at ED	Plasma and urine MR-proADM	To study the usefulness of MR-proADM as a biomarker of acute and chronic renal parenchyma damage in fUTI.	Culture, DMSA	0.92	0.66 nmol/L (P-MR-proADM, for predicting RS)	83.3	81.8	-	P-MR-ProADM appears to have prognostic utility as a predictor of RS.
**Febrile Neutropoenia**											
Demirkaya et al. (2015) [28]	P, O, SC	<18 yr 50 episodes of fever in 37 neutropenic oncology patients	ADM PCT CRP	To determine differences in serum ADM in febrile neutropoenia patient categories (CDI, FUO, MDI, sepsis).	Radiological imaging, cultures, clinical signs, Goldstein 2005 sepsis definitions	In the MDI group, ADM levels on day 3 were significantly higher than those in the CDI and FUO groups.					A correlation between serum adrenomedullin levels and the severity of febrile neutropoenia could not be demonstrated. Among adrenomedullin, CRP, and PCT, PCT demonstrates the highest correlation with the severity of infection.
Kesik V. et al. (2016) [29]	P, O, SC	10 yr (1.66–16) * 36 febrile episodes in 14 neutropenic children with solid tumours	ADM	To evaluate the role of ADM in predicting the prognosis for patients with FN.	Culture Risk categories, as described by Alexander et al., 2002	0.76 0.628	263.5 ng/L (High-risk patients) 505.5 ng/L (Culture positivity)	88 85.7	60 50	84.6/66.7 30/93	ADM levels at admission were useful in identifying those at high risk and culture positivity in patients with solid tumours.
Agnello L. et al. (2020) [30]	P, O, SC	10 yr (0–17) * 36 febrile neutropenic episodes in 26 children being treated by oncology	PSP MR-proADM	To evaluate MR-proADM and PSP plasma levels and their kinetics in paediatric oncology patients with FN to assess their usefulness in the management of these patients.	Culture and clinical signs	0.62	.	-	-	-	MR-proADM had low diagnostic accuracy for blood culture positivity.
Fawsi M.M. et al. (2021) [31]	P, O, SC	1–15 yr 100 neutropenic patients due to hematologic malignancies	MR-proADM CRP	To assess the utility of MR-proADM as an early marker for sepsis in severely neutropenic patients with hematologic malignancies.	Culture, Goldstein 2005 sepsis criteria	0.939	2.4 nmol/L (bacteraemia/sepsis discrimination)	91.6	85.1	83.3/92.4	MR-proADM is a promising early marker for sepsis in severely neutropenic young patients with hematologic malignancies.
**Other Infections**											
Michels M. et al. (2011) [32]	CC, SC	<15 yr *n* = 88 (43 children with DHF, 28 children with DSS, 17 controls)	MR-proADM CoPEP	To determine the relationship of MR-proADM and CoPEP to outcomes and their potential as biomarkers of plasma leakage in children with DHF and DSS.	WHO criteria, low albumin, pleural effusion	Plasma MR-proADM concentrations were significantly higher in the DHF and DSS groups at enrolment than in the healthy controls. MR-proADM concentrations in the DHF group had decreased by day 2, remaining elevated in the DSS group. Inverse correlation between MR-proADM and serum albumin at enrolment. MR-proADM concentrations were positively correlated with the pleural effusion index at day 2.					MR-proADM may have a functional role in limiting endothelial hyperpermeability during DHF/DSS. MR-proADM may be a candidate biomarker to predict the development of DHF/DSS.
Benito, J. et al. (2013) [33]	P, O, MC (7 ED)	1–36 mo *n* = 1035 ED well-appearing children with FWS	MR-proADM CT-pro-ET-1	To assess the usefulness of MR-pro-ADM and CT-pro-ET-1 in predicting BI and IBI in well-appearing infants with FWS.	PCT	0.59 0.63	≥0.6 nmol/L (predicting BI) 0.7 nmol/L (predicting IBI)	- -	- -	- -	The overall performance of MR-proADM as a diagnostic marker of BI and IBI is poor.
Bueno-Campaña, M. et al. (2018) [34]	P, O, MC (N = 2)	< 6 mo *n* = 33 Children with bronchiolitis	MR-proADM	To explore the relationship between the need for respiratory support and MR-proADM.	-	Children who needed nasal cPAP or MV presented higher MR-proADM levels than the group that required just high-flow therapy or no extra support (*p* < 0.05).					MR-proADM could be a potential biomarker for the severity of acute bronchiolitis.
Girona-Alarcón, M. (2021) [35]	P, O, SC	Adults/children *n* = 24 (20 adults with ARDS, 4 children with MIS-C)	MR-proADM PCT CRP Troponins NT-proBNP	To describe the characteristics of the disease in each specific population and to analyse the differences between adults and children.	-	MR-proADM levels were higher in children than in adults: 1.72 vs. 0.78 nmol/L (*p* = 0.017).					MR-proADM use should be studied in larger samples, since it could be helpful to pinpoint the risk of MIS-C in children infected with COVID.

The recorded results correspond to adrenomedullin results. Sensitivity, specificity and positive/negative predictive values are expressed as percentages. * Median and interquartile range. AUC: area under curve; Se: sensitivity; Sp: specificity; PPV: positive predictive value; NPV: negative predictive value; P: prospective; O: observational; MC: multicentre; PICU: paediatric intensive care unit; yr: years; MR-proADM: mid-regional pro-adrenomedullin; CT-proET-1: carboxy-terminal pro-endothelin-1; PCT: procalcitonin; CRP: C-reactive protein; PRISMIII: Pediatric Risk of Mortality III score; PIM: Pediatric Index of Mortality; SC: single centre; d: days; CC: case control; SIRS: systemic inflammatory response syndrome; ED: emergency department; AA: acute appendicitis; WBC: white blood cell count; PAS: pediatric appendicitis score; CAP: community-acquired pneumonia; pro-ADM: pro-adrenomedullin; CoPEP: copeptin; mo: months; sTREM-1: soluble triggering receptor expressed on myeloid cells 1; MR-proANP: mid-regional pro-atrial natriuretic peptide; BTS: British Thoracic Society; IL-1β: Interleukin 1β; w: weeks; UTI: urinary tract infection; ADM: adrenomedullin; APN: acute pyelonephritis; DMSA: 99mTc dimercaptosuccinic acid scintigraphy; PSP: presepsin; TREM-1: triggering receptor expressed on myeloid cells 1; ESR: erythrocyte sedimentation rate; fUTI: febrile urinary tract infection; RS: renal scarring; CDI: clinically documented infection; FUO: fever of unknown origin; MDI: microbiological documented infection; FN: febrile neutropoenia; DHF: dengue haemorrhagic fever; DSS: dengue shock syndrome; WHO: world health organization; FWS: fever without source; BI: bacterial infection; IBI: invasive bacterial infection; cPAP: continuous positive airway pressure; MV: mechanical ventilation; ARDS: acute respiratory distress syndrome; MIS-C severe multisystem inflammatory syndrome, COVID-19 related; NT-proBNP: N-terminal pro-B-type natriuretic peptide.

**Table 2 children-09-01181-t002:** Adrenomedullin and cardiovascular disease.

Scheme	Type of Study	Setting	Biomarkers Studied	Aim	Gold Standard	AUC	Cut-Off	Se	Sp	PPV/ NPV	Conclusion
**Congenital Heart Disease**											
Yoshibayashi M. et al. (1999) [36]	CC, SC	0.8–18 yr *n* = 28 (16 cyanotic w/CHD, 12 KD patients as controls)	ADM	To investigate the pathophysiological significance of ADM in hypoxaemia caused by cyanotic CHD.	-	Patients with cyanotic CHD showed significantly higher concentrations of ADM and an increased uptake of adrenomedullin in the pulmonary circulation (compared to arterial-venous levels of ADM) was also detected.					ADM levels may function as a compensatory mechanism for hypoxaemia in cyanotic congenital heart disease.
Watanabe K. et al. (2003) [37]	CC, SC	8.3 ± 7.2 yr * *n* = 89 82 children with CHD, 7 healthy controls	ADM-m ADM-Gly	To investigate the pathophysiological role of two forms of ADM, ADM-m and ADM-Gly, in CHD.	Echocardiography and cardiac catheterisation	Plasma ADM-m and ADM-Gly were examined in cyanotic CHD and intracardiac repair with PH. They were negatively correlated with SA_sat_ and mixed venous oxygen saturation, and positively correlated with Rp. In the multiple regression analysis, SA_sat_ and Rp were independently correlated with ADM. Venous ADM-m levels were significantly higher than arterial ADM-m levels.					ADM levels may play beneficial roles in reducing pulmonary arterial resistance or alleviating hypoxaemia in these patients. Results also suggest that the mature form is extracted in pulmonary circulation.
Zhu X.B. et al. (2006) [38]	CC, SC	*n* = 48 (42 children with CHD, 6 recovered KD patients as controls)	ADM	To investigate the pathophysiological role of ADM in CHD.	Echocardiography and cardiac catheterisation	Plasma ADM levels are increased in congenital heart disease with high pulmonary blood flow and hypertension or with cyanosis, *p* < 0.01. Plasma ADM levels are positively correlated with pulmonary vascular resistance (r = 0.406, *p* < 0.01).					Increased ADM levels may play a role in reducing the pulmonary arterial resistance and alleviating hypoxaemia in these patients.
**Cardiovascular Surgery**											
Komai H. et al. (1998) [39]	P, O, SC	2–8 mo children *n* = 21 - 6 children with cyanotic CHD - 8 children with high pulmonary arterial flow due to CHD - 7 adults with MV disease	ADM	To evaluate ADM in patients undergoing CPB as a marker of pulmonary vascular damage.	-	The plasma ADM level increased significantly after CPB in each group. The high pulmonary arterial flow group had significantly lower ADM levels before and after CPB compared to other groups. In this group, there was a significant negative correlation between pre-CPB mPAP and ADM levels after CPB.					ADM may provide information regarding CPB-induced endothelial damage. The difference in ADM production in the high pulmonary arterial flow group may have been a consequence of the pre-existing pulmonary damage in these patients.
Szekely L. et al. (2000) [40]	P, O, SC	1–99 mo *n* = 19 Children undergoing surgical CHD repair	ADM	To study whether perioperative myocardial injury could be altered by the presence of ADM.	Troponin-I	Preoperative ADM levels in the group with little or no evidence of myocardial injury after the surgery were significantly greater than the groups with either moderate or severe injury.					Higher preoperative ADM levels are associated with lower levels of myocardial injury (as assessed by troponin-I release) during surgery for congenital heart defects.
Takeuchi M. et al. (2001) [41]	P, O, SC	7 mo–6 yr *n* = 13 Children with CHD during CPB	ADM VP ANP BNP ALD	To elucidate the effects of ADM on fluid homeostasis during CPB.	-	ADM levels increased gradually, with a peak 60 min after CPB, and decreased 24 h after the operation. ADM levels were correlated with urine volume and with brain natriuretic peptide during CPB.					ADM plays an important role in fluid homeostasis during CBP, in cooperation with other hormones.
Florio P. et al. (2008) [42]	P, O, SC	126 ± 110 d * *n* = 50 Infants with CHD undergoing CPB	ADM	To determine whether ADM measurement is useful for monitoring cerebral distress during CPB.	Neurological signs at physical examination using the Amiel–Tison test	0.897	17.4 ng/L (predicting brain damage)	100	73	-	Infants who developed abnormal neurologic sequelae had significantly higher MCA PI values and lower ADM concentrations; thus, these indicators may be useful for the early identification of infants at risk for brain damage.
Abella R. et al. (2012) [43]	P, O, SC	0–9 mo *n* = 48 Children undergoing CPB	ADM	To investigate whether perioperative ADM levels can predict risk of LCOS.	Clinical/laboratory findings, inotropic score, cardiac death, echocardiography	0.842	27 pg/L (predicting LCOS)	100	64.1	39.1/100	ADM might be, alone or in combination with standard parameters, a promising predictor of LCOS in infants subjected to open-heart surgery with CPB.
Arkader R. et al. (2013) [44]	P, O, SC	39 ± 16 mo * *n* = 19 Children undergoing CPB and receiving 30 mg/kg MP after anaesthesia	C-peptide CRP IL-6 ADM	To improve our understanding of the metabolic and inflammatory factors that are involved in glucose regulation in children after CPB.	-	The ADM levels before CPB were slightly higher than normal, and increases of 323% were observed on day 1 after CPB; levels returned to baseline on day 3. ADM was correlated to insulin variations and was an independent factor associated with lower insulin concentrations in the multiple regression model.					ADM may be a predictor of low insulin concentration after CPB.
Pérez-Navero J. et al. (2017) [45]	P, O, SC	10 d–15 yr *n* = 117 Children after CPB	ANP BNP CoPEP MR-proADM cTn-I	To assess biomarkers as indicators of LCOS in children undergoing CPB.	Echocardiography (LVEF), PiCCO (CI), clinical and analytical criteria	0.848	1.5 nmol/L 24 h post-CPB (predicting LCOS)	88	66	50/93	cTn-I at 2 h post-CPB and ADM at 24 h post-CPB were independent predictors of LCOS.
Pérez-Navero J. et al. (2018) [46]	P, O, SC	10 d–15 yr *n* = 117 Children after CPB	MR-proADM cTn-I	To determine the predictive value of IS, VIS, MR-proADM, and cTn-I for LCOS in children undergoing CPB.	Echocardiography (LVEF), PiCCO (CI), clinical and analytical criteria	0.81 0.850	Prediction model 1 (age, CPB > 120 min, VIS) Prediction model 2 (model 1 + cTn-I, MR-proADM)	55.5 61.3	92.9 5.7	74.7/79.59 61.3/85.7	The VIS score at 2 h post-CPB was identified as an independent early predictor of LCOS. This predictive value was not significantly increased when associated with cardiac biomarkers for LCOS.
Bobillo-Pérez S. et al. (2019) [47]	P, O, SC	1 mo–16 yr *n* = 111 Children after CPB	PCT MR-proADM pro-ANP	To assess the usefulness of PCT, pro-ADM, and pro-ANP as predictors of need for MV, inotropic support, and bacterial infection in patients after CPB.	-	0.721 0.738	1.22 nmol/L immediately after CPB (predicting MV need) 1.01 nmol/L 24–36 h after CPB (predicting MV need)	69.5 72.9	87 64.8	80.2/78.9 51.9/82.2	Pro-ADM and pro-ANP are good predictors of need for MV and LOS after CPB. Procalcitonin is useful for predicting bacterial infection.
Bobillo-Pérez S. et al. (2020) [48]	P, O, SC	2.1 yr (0.6–6.6) ** *n* = 113 Children undergoing cardiac surgery	MRpro-ADM pro-ANP	To evaluate the utility of pro-ANP and pro-ADM levels prior to CPB for predicting the need for intensive post-CPB support.	-	0.724 0.855	pro-ADM for predicting increased respiratory support pro-ADM for predicting increased inotropic support	-	-	-	In the multivariable analysis, pro-ADM wasn’t identified as an independent predictor for increased need for respiratory or inotropic support.
**Fontan Surgery**											
Hiramatsu T. et al. (1999) [49]	CC, SC	1–14 yr *n* = 16 8 Fontan procedures, 8 biventric. repair with normal CVP	ET-1 ADM	To examine the time course of ET-1 and ADM and to explore their influence on pulmonary vascular tone.	Cardiac catheterisation	ET-1 and ADM increased after CPB in both groups. However, plasma ET-1 levels were significantly elevated and ADM levels were significantly lower at 6 and 24 h after CPB in the Fontan group when compared with the control group.					An imbalance between increased ET-1 and decreased ADM after CPB during the Fontan procedure induces vasoconstriction.
Watanabe K. et al. (2007) [50]	CC, SC	1.4–22.6 yr *n* = 40 29 Fontan procedures, 11 recovered KD	ADM-m ADM-Gly ADM-T	To investigate the significance of molecular forms of ADM in patients after the Fontan procedure.	Follow-up cardiac catheterisation (period between the Fontan procedure and the examination was 5.8 ± 4.9 yr)	Fontan patients had significantly higher venous concentrations of ADM-T, ADM-Gly, and ADM-m than controls. There was a significant difference in ADM-m levels in the pulmonary artery and femoral artery. The venous concentration of ADM-m correlated negatively with cardiac output.					ADM may be involved in the regulation of pulmonary arterial tone following Fontan surgery.
Kaiser R. et al. (2014) [51]	CC, SC	4–36 yr *n* = 65 53 after Fontan procedure, 12 healthy subjects	MR-proADM	To assess the utility of MR-proADM as a predictor of Fontan procedure failure.	Echocardiography, abdominal and pleural ultrasound, NYHA classification	0.985	>0.520 nmol/L (predicting Fontan failure)	100	93.9	57.1/100	Serial measurements of MR-proADM levels may help identify patients at risk for a failing Fontan circulation, especially when these exceed 0.520 nmol/L.
**Heart Failure**											
Randa Abdel Kader M. et al. (2007) [52]	CC, SC	*n* = 84 38 adults w/CHF 21 children w/CHF 15 adult and 10 paediatric healthy controls	ADM ANP	To evaluate ADM and ANP in patients with CHF and investigate their relationship with haemodynamic variables.	NYHA functional classification, echocardiography	Plasma levels of ADM and ANP increased in adult and paediatric patients with CHF, irrespective of the cause. They were positively correlated with each other and negatively correlated with LVEF and FS.					ADM and ANP may be used to identify high-risk subjects for HF during more invasive procedures.
Hauser J. et al. (2016) [53]	CC, MC (N = 2)	0–24 yr *n* = 203 114 patients w/HF, 89 controls	MR-proANP sST2 GDF-15 MR-proADM NT-proBNP	To assess the diagnostic utility of four novel biomarkers in paediatric HF.	Presence of HF symptoms, abnormal systolic ventricular function via MRI or echocardiography	ROC analysis showed poor accuracy for MR-proADM (AUC 0.61, 95% CI 0.48–0.69, *p* = 0.029). In a subset of patients with DCM, MR-proADM was associated with poor LV function, but showed poor accuracy for its diagnosis.					MR-proADM shows unsatisfactory diagnostic power.
**Dilated Cardiomyopathy**											
Kılınc M. et al. (2003) [54]	CC, SC	5 mo–14 yr *n* = 21 11 DCM, 10 healthy controls	Plasma and urine ADM and NO	To determine plasma and urine AM and NO in children with idiopathic DCM and correlate these with other clinical and laboratory findings.	Echocardiography, signs and symptoms of DCM	Plasma and urine ADM levels were significantly lower than in the healthy controls. Plasma and urine ADM levels were negatively correlated with EF and FS.					Low ADM levels may be a bad prognostic factor for children with DCM in advanced stages.
**Orthostatic Tachycardia**											
Zhang F. et al. (2012) [55]	CC, SC	7–14 yr *n* = 77 57 children with POTS, 20 healthy children as controls	MR-proADM	To explore the predictive value of MR-proADM in assessing the therapeutic efficacy of midodrine hydrochloride for children with POTS.	Symptom scoring and HUT/HUTT test	0.879	61.5 pg/mL (predicting efficacy of midodrine hydrochloride)	100	71.6	-/-	MR-proADM can help guide midodrine hydrochloride therapy in the management of POTS in children, identifying those who will have a good response to the drug.
Li H. et al. (2015) [56]	P, O, SC	14.5 ± 4.5 yr * *n* = 53 children with POTS	MR-proADM	To explore the predictive value of baseline plasma MR-proADM for the long-term survival of children with POTS treated with midodrine hydrochloride.	Orthostatic intolerance symptom score and symptom-free survival	At the 60-month follow-up, patients with baseline MR-proADM of >61.5 ng/L had a significantly lower symptom score. At the 72-month follow-up, the symptom score was similar, while symptom-free survival was significantly higher if baseline MR-proADM was >61.5 ng/L.					The baseline plasma MR-proADM level is valuable for predicting the long-term survival of children with POTS treated with midodrine hydrochloride.
**Transcatheter PDA Closure**											
Wu R.Z. et al. (2010) [57]	CC, SC	*n* = 115 55 children w/transcatheter PDA closure, 60 normal children	ADM BNP	To observe changes in ADM and BNP before and after transcatheter closure in children with PDA.	-	Before transcatheter closure, concentrations of plasma ADM were significantly higher in patients with PDA compared to the control group. Plasma ADM was significantly reduced at day 3 and month 3 after transcatheter closure.					Plasma ADM levels decreased significantly after transcatheter closure in children with PDA.

The recorded results correspond to adrenomedullin. Sensitivity, specificity, and positive/negative predictive values are expressed as percentages.* Mean and standard deviation. ** Median and interquartile range. AUC: area under curve; Se: sensitivity; Sp: specificity; PPV: positive predictive value; NPV: negative predictive value; CC: case control; SC: single centre; yr: years; CHD: congenital heart disease; KD: Kawasaki disease; ADM: adrenomedullin; ADM-m: mature adrenomedullin; ADM-Gly: glycine-extended adrenomedullin; PH: pulmonary hypertension; SA_sat_: systemic arterial oxygen saturation; Rp: pulmonary arterial resistance; P: prospective; O: observational; mo: months; MV: mitral valve; CPB: cardiopulmonary bypass; mPAP: mean pulmonary arterial pressure; VP: vasopressin; ANP: atrial natriuretic peptide; BNP: brain natriuretic peptide; ALD: aldosterone; d: days; MCA PI: middle cerebral artery pulsatility index; LCOS: low cardiac output syndrome; MP: methylprednisolone; IL-6: interleukin 6; CoPEP: copeptin; MR-proADM: mid-regional pro-adrenomedullin; cTn-I: cardiac troponin I; LVEF: left ventricle ejection fraction; CI: cardiac index; IS: inotropic score; VIS: vasoactive-inotropic score; PCT: procalcitonin; pro-ANP: pro atrial natriuretic peptide; MV: mechanical ventilation; LOS: length of stay; CVP: central venous pressure; ET-1: endothelin-1; ADM-T: total adrenomedullin; CHF: congestive heart failure; ANP: atrial natriuretic peptide; NYHA: New York Heart Association; FS: fractional shortening; HF: heart failure; MC: multicentre; sST2: soluble ST2; GDF-15: growth differentiation factor-15; NT-proBNP: N-terminal pro-B natriuretic peptide; MRI: magnetic resonance imaging; DCM: dilated cardiomyopathy: LV: left ventricular; NO: nitric oxide; EF: ejection fraction; POTS: postural orthostatic tachycardia syndrome; HUT: head-up test; HUTT: head-up tilt test; PDA: patent ductus arteriosus; BNP: B-type natriuretic peptide.

**Table 3 children-09-01181-t003:** Adrenomedullin and pulmonary hypertension.

Study	Type of Study	Setting	Biomarkers Studied	Aim	Gold Standard	AUC	Cut-Off	Se	Sp	PPV/ NPV	Conclusion
Yoshibayashi M. et al. (1997) [58]	CC, SC	9 mo–19 yr *n* = 51 25 C-PH, 10 P-PH, 16 no PH	ADM-LI	To elucidate the pathophysiological significance of ADM in PH.	Cardiac catheterisation	Plasma AM-LI concentrations in the C-PH group and the P-PH group were significantly higher than in the no PH group. There is a significant decrease in the plasma AM-LI concentration between the PA and PV. There is a significant correlation between plasma AM-LI in the PA and pulmonary artery pressure.					ADM may be involved in the cardiovascular regulation or homeostasis of pulmonary circulation in pulmonary hypertension.
Vijay P. et al. (1998) [59]	P, O, SC	9 yr *n* = 30 Patients w/CHD	ADM ET-1 NO_2_ NO_3_	To examine the influence of these biomarkers in the development of postoperative PH.	Cardiac catheterisation or echocardiography	ADM levels were significantly higher in LF groups compared to HF groups (*p* < 0.0001). Upon initiating CPB, ADM levels in the LF group declined from their preoperative value to a level similar to that seen in the HF group. The levels of ADM gradually increased during the first 12 h post-op in all groups (*p* ≤ 0.05)					ADM appears to affect baseline vascular tone in patients with intact endothelial function.
Nakayama, T. (2001) [60]	P, O, SC	12 ± 4 yr * *n* = 17 Children w/P-PH	ANP BNP ET-1 ADM	To investigate whether plasma levels of ADM are useful for assessing the severity of P-PH.	NYHA classification, pulmonary haemodynamics, and 6 min. walk test	ADM significantly decreased at 1 month and at 3 months. There was a significant relationship between the changes in adrenomedullin at 3 months compared to values upon initiating PGI2 therapy and the changes in mean pulmonary arterial pressure.					ADM was valuable for evaluating both cardiac performance and pulmonary haemodynamics after long-term treatment with PGI2 in patients with primary pulmonary hypertension.
Lu H. et al. (2003) [61]	CC, SC	2 mo–16 yr *n* = 48 33 patients w/CHD, 15 healthy children	ADM ET-1 NO	To investigate their role in CHD with PH.	-	Plasma ADM levels were significantly higher in patients with CHD than in the control group (*p* < 0.05) and increased significantly as PH severity increased from non-PH to mild to moderate-severe (*p* < 0.05, *p* < 0.01). On the 7th day after CPB, plasma ADM levels in the PH group were significantly decreased in comparison with those before the operation. Plasma ADM levels in CHD were positively correlated with PASP.					ADM may play an important role in the development of PH in patients with CHD. ADM may be involved in the defence mechanism against further increases in pulmonary arterial pressure.
Wang, T. (2005) [62]	P, O, SC	*n* = 52 children with CHD	ADM UII	To evaluate the effects and clinical significance of ADM and UII as regards PH.	-	As pulmonary hypertension increases in severity, the plasma levels of ADM increase. There is a positive correlation between PSAP and plasma ADM levels. Plasma levels of ADM in each group after CPB is lower than that of each group before the operation.					Measuring the levels of ADM may be a reliable method to monitor changes in pulmonary pressure and the worsening of pulmonary hypertension.

The recorded results correspond to adrenomedullin. Sensitivity, specificity, and positive/negative predictive values are expressed as percentages. * Mean and standard deviation. AUC: area under curve; Se: sensitivity; Sp: specificity; PPV: positive predictive value; NPV: negative predictive value; CC: case–control; SC: single centre; mo: months; yr: years; C-PH: cardiac pulmonary hypertension; P-PH: primary pulmonary hypertension; PH: pulmonary hypertension; ADM-LI: plasma adrenomedullin-like immunoreactivity; ADM: adrenomedullin; PA: pulmonary artery; PV: pulmonary vein; P: prospective; O: observational; CHD: congenital heart disease; ET-1: endothelin-1; NO_2_: nitrites; NO_3_: nitrates; LF: low pulmonary flow; HF: high pulmonary flow; CPB: cardiopulmonary bypass; ANP: atrial natriuretic peptide; BNP: brain natriuretic peptide; NYHA: New York Heart Association; PGI2: prostaglandin I2; mo: months; NO: nitric oxide; PASP: pulmonary artery systolic pressure; UII: urotensin-II.

**Table 4 children-09-01181-t004:** Adrenomedullin in nephro-urological disease.

Study	Type of Study	Setting	Biomarkers Studied	Aim	Gold Standard	AUC	Cut-Off	Se	Sp	PPV/ NPV	Conclusion
Balat et al. (2000) [63]	CC, SC	6 mo–10 yr *n* = 25 10 w/Bartter syndrome, 5 w/pseudo-Bartter, 10 healthy controls	Urine and plasma total nitrite and ADM	To verify whether NO and ADM play a role in the reduced vascular response seen in Bartter syndrome.	-	Plasma ADM levels were higher in those with Bartter syndrome than in the other groups (*p* > 0.05). Urine ADM was increased in children with Bartter syndrome and pseudo-Bartter syndrome as compared with healthy controls during the pre-treatment period (*p* < 0.05) and in Bartter syndrome compared to pseudo-Bartter syndrome after 6 month of therapy (*p* < 0.05).					The increased production of ADM may have a role in the reduced vascular response seen in Bartter syndrome.
Balat A. et al. (2000) [64]	CC, SC	2–10 yr *n* = 23 13 w/clinical MCNS, 10 healthy controls	Urinary and plasma total nitrite and ADM	To study plasma and urine ADM and NO concentrations in children w/MCNS during relapse and remission.	Clinical criteria	Plasma ADM concentrations during relapse were significantly lower than in remission and controls, and urine ADM levels were significantly higher in relapse than in remission and controls.					The important changes in plasma and urine ADM levels in relapse suggested that these changes may be compensatory when the body experiences severe proteinuria.
Balat A. et al. (2002) [65]	CC, SC	8.05 ± 1.61 yr * *n* = 41 30 children w/PNE, 18 healthy controls	Urinary and plasma total nitrite and ADM	To investigate if plasma and urine ADM and nitrite levels are altered in children with PNE.	Clinical criteria	Plasma and urine ADM levels were significantly lower in children with PNE than in controls (*p* < 0.001 and *p* < 0.01). There was a negative correlation between 24-h urinary K+ excretion and ADM levels (r = −0.47, *p* < 0.01). No correlation was found between the number of bedwetting episodes per week/duration of enuresis with urinary ADM levels.					Decreased ADM levels may be a compensatory response when there is abnormal electrolyte and water excretion.
Balat A. et al. (2003) [66]	CC, SC	6.43 ± 2.62 yr *n* = 20 14 patients w/DI, 6 patients w/VUR as controls	Tissue levels of NO and ADM	To determine the tissue levels of NO and ADM in children with DI and compare them with children with normal bladder activity.	Urodynamic testing	ADM levels measured via bladder biopsy (pmol/g tissue) were increased in children with DI (*p* < 0.001).					Increased ADM appears to be compensatory for decreased NO production in the smooth muscle of the bladder in DI patients.
Kalman S. et al. (2005) [67]	CC, SC	1–13 yr *n* = 74 w/recurrent UTI: 25 w/RPS and w/VUR, 16 w/RPS and w/o VUR, 12 w/o RPS and w/VUR, 21 healthy children	Plasma and urine ADM	To determine plasma and urine ADM levels in children with RPS and VUR.	Cystoureterography DMSA	Plasma ADM levels were not significantly higher in the control group than in the rest of the groups (*p* = 0.162). Urine ADM levels were higher in the control group than in the three study groups (*p* < 0.005).					Urine ADM can be a prognostic factor in the long-term follow-up of paediatric cases with VUR.

The recorded results correspond to adrenomedullin. Sensitivity, specificity, and positive/negative predictive values are expressed as percentages. * Mean and standard deviation. AUC: area under curve; Se: sensitivity; Sp: specificity; PPV: positive predictive value; NPV: negative predictive value; CC: case–control; SC: single centre; yr: years; ADM: adrenomedullin; DMSA: 99mTc dimercaptosuccinic acid scintigraphy; mo: months; NO: nitric oxide; MCNS: minimal change nephrotic syndrome; PNE: primary nocturnal enuresis; DI: detrusor instability; RPS: renal parenchymal scar; VUR: vesicoureteral reflux.

**Table 5 children-09-01181-t005:** Adrenomedullin and endocrine disease.

Study	Type of Study	Setting	Biomarkers Studied	Aim	Gold Standard	AUC	Cut-Off	Se	Sp	PPV/ NPV	Conclusion
Del Ry S. et al. (2016) [68]	CC, SC	12.5 ± 0.4 yr * *n* = 83 51 obese adolescents, 32 healthy adolescents	MR-proADM	To assess plasma MR-proADM levels in obese adolescents compared to normal-weight subjects.	National BMI reference data specific for age and sex	Plasma MR-proADM levels were significantly higher in obese adolescents than in normal-weight ones (*p* < 0.0001). MR-proADM correlated with BMI z-score, fat mass, circulating insulin, HOMA-IR, total cholesterol, and LDL cholesterol. Fat mass and BMI z-score were independent determinants of circulating MR-proADM.					Obese adolescents have higher circulating levels of MR-proADM compared to those of normal weight, suggesting its important involvement in obese patients.
Metwalley K.A. et al. (2018) [69]	CC, SC	9.76 ± 2.21 yr * *n* = 120 60 obese children, 60 non-obese children	ADM	To determine the plasma levels of ADM in obese children and their relationship to LV function.	Echocardiography	-	52 pg/mL (predicting LV hypertrophy)	94.3	92.5	-	Measuring plasma ADM levels in obese children may help to identify those at high risk of developing LV hypertrophy and dysfunction.
El-Habashy S. et al. (2010) [70]	CC, SC	13.9 ± 3.2 yr * *n* = 85 55 children w/T1DM, 30 healthy controls	ADM HbA1c	To assess ADM levels in children and adolescents with T1DM and their correlation with diabetic MVC.	Indirect ophthalmoscope examination of the fundus	ADM levels were significantly increased in patients with and without MVC compared to the control group, with higher levels in those with MVC. Plasma ADM levels were positively correlated with both duration of diabetes and HbA1c.					ADM may have a role in the diabetic vasculopathy of children and adolescents with T1DM.
Semeran K. et al. (2013) [71]	CC, SC	8–18 yr *n* = 62 41 patients with T1DM, 21 healthy controls	HbA1c Lipid profile IL-17 VEGF ADM	To assess the use of biomarkers in patients with T1DM with no visible lesions for predicting retinal dysfunction.	S-cone ERG protocol	A statistically significant finding was that the ADM level in the diabetes group was lower than in the control group (*p* < 0.05).					The changes observed in the ADM levels support its possible involvement in the microvascular complications of diabetes. Different-than-expected study results as regards ADM concentration indicate that further studies are needed.

The recorded results correspond to adrenomedullin. Sensitivity, specificity, and positive/negative predictive values are expressed as percentages. * Mean and standard deviation. AUC: area under curve; Se: sensitivity; Sp: specificity; PPV: positive predictive value; NPV: negative predictive value; CC: case–control; SC: single centre; yr: years; MR-proADM: mid-regional pro-adrenomedullin; BMI: body mass index; ADM: adrenomedullin; LV: left ventricle; T1DM: type 1 diabetes mellitus; ADM: adrenomedullin; MVC: microvascular complications; IL-17: interleukin-17; VEGF: vascular endothelial growth factor.

**Table 6 children-09-01181-t006:** Adrenomedullin and rheumatic diseases.

Study	Type of Study	Setting	Biomarkers Studied	Aim	Gold Standard	AUC	Cut-Off	Se	Sp	PPV/ NPV	Conclusion
Nishida K. et al. (2001) [72]	P, O, SC	0.4–2.6 yr *n* = 6 Children with acute KD	ADM	To test ADM as an early detection biomarker of coronary artery vasculitis in KD.	Echocardiography	ADM levels were markedly elevated before treatment, especially in patients with coronary artery dilation. After treatment, plasma adrenomedullin levels substantially decreased.					ADM may be useful to monitor Kawasaki disease patients during its acute phase and may help to diagnose coronary artery involvement.
Islek I. et al. (2003) [73]	CC, SC	11.4 ± 3.1 * *n* = 28 16 children with HSP, 12 healthy controls	Plasma and urine NO and ADM	To measure ADM and NO levels in children with HSP.	Diagnostic criteria	Plasma and urine ADM levels were significantly higher in the acute phase of HSP than in the controls. No difference was seen between remission phase levels and control ADM levels.					ADM may have a role in the immune-inflammatory process of HSP, especially in the active stage.
Balat A. et al. (2005) [54]	CC, SC	7–14 yr *n* = 25 11 children with ARF, 14 healthy controls	Plasma and urine NO and AM	To investigate whether an association between levels of ADM/NO and ARF exists.	Diagnostic criteria	Plasma and urine ADM levels were significantly higher in children with ARF, irrespective of whether they were in the acute or convalescent phases. Plasma and urine levels of ADM in the acute phase showed positive correlation with ESR and negative correlation with EF.					ADM may play a role in the immune-inflammatory process of ARF. However, increased levels may also be the result of inflammatory injury in ARF.
Balat A. et al. (2006) [74]	CC, SC	3–16 yr *n* = 15 w/FMF, 15 healthy controls	Plasma and urine NO and ADM	To determine the levels of ADM and NO in children w/FMF and compare with the healthy controls.	Diagnostic criteria	Plasma and urine levels were significantly higher in FMF patients than in controls.					ADM may have a role in the immuno-inflammatory process of FMF, although whether these act to promote or protect against further inflammatory injury is not clear.
Kalman S. et al. (2012) [75]	CC, SC	9.2 ± 4.7 yr * *n* = 37 (16 homozygous M694V, 10 heterozygous M694V, 11 other mutations)	Urine and plasma ADM	To compare plasma and urine ADM levels of FMF patients who exhibit M694V homozygosity and patients with other genotypes.	-	Plasma ADM levels were higher in FMF patients w/homozygous M694V than other patients (heterozygous M694V and other mutations). No differences between heterozygous M694V and other mutation groups, nor in urine ADM levels.					Although the results are favourable, more studies are needed to demonstrate the association between homozygous M694V mutation and ADM levels.
Polat A. (2015) [76]	CT, SC	7.8 ± 2 yr *n* = 37 Children with confirmed FMF	ADM	To investigate ADM as a marker for inflammation in paediatric patients with FMF who are using colchicine at different doses.	ESR and CRP levels	ADM levels were similar in all visits (*p* = 0.954) and did not show any differences between the first 3 visits and following 3 visits i.e., before and after changing the dosage.					No alterations in ADM levels were demonstrated at any visits, which may suggest the continuation of subclinical inflammation in these patients.

The recorded results correspond to adrenomedullin. Sensitivity, specificity, and positive/negative predictive values are expressed as percentages. * Mean and standard deviation. AUC: area under curve; Se: sensitivity; Sp: specificity; PPV: positive predictive value; NPV: negative predictive value; P: prospective; O: observational; SC: single centre; yr: years: KD: Kawasaki disease; ADM: adrenomedullin; CC: case–control; HSP: Henoch-Schönlein purpura; NO: nitric oxide; ARF: acute rheumatic fever; ESR: erythrocyte sedimentation rate; EF: ejection fraction; FMF: Familial Mediterranean Fever; CT: clinical trial; CRP: C-reactive protein.

**Table 7 children-09-01181-t007:** Other articles published on adrenomedullin in children.

Study	Type of Study	Setting	Biomarkers Studied	Aim	Gold Standard	AUC	Cut-Off	Se	Sp	PPV/ NPV	Conclusion
Özgür B.G. et al. (2017) [77]	CC, SC	9.2 ± 2.8 yr * *n* = 90 45 children with ADHD, 45 healthy children	ADM NO	To compare plasma ADM and NO levels of newly diagnosed, treatment-naive patients with ADHD and healthy children.	K-SADS-PL-T schedule	There were no statistically significant differences in NO and ADM levels, neither between the groups nor ADHD subtypes.					The role of ADM in the pathophysiology of ADHD could not be demonstrated.
Zoroglu S.S. et al. (2003) [78]	CC, SC	2–12 yr *n* = 48 26 autistic patients, 22 healthy controls	ADM NO	To assess the role of NO and ADM in autism.	DSM-IV diagnostic criteria	The mean values of plasma total nitrite and ADM levels in the autistic group were significantly higher than control values.					ADM may have a pathophysiological role in autism; this subject requires much further research.
Kucukosmanoglu E. et al. (2012) [79]	CC, SC	5–15 yr *n* = 47 27 children with acute asthma attack, 20 controls	ADM	To determine changes in ADM levels during an acute asthma attack and its association with allergen sensitivity.	GINA classification, prick tests	No significant differences were found in ADM levels between the controls and patients in either the acute attack or remission period. ADM levels were higher in the acute attack period in those patients with a severe attack, patients sensitive to an allergen, and patients with history of atopic dermatitis.					ADM may play a role in children with atopic dermatitis and may also have a role in the immuno-inflammatory process of asthma.
Piccin A. et al. (2015) [80]	CC, SC	*n* = 128 111 children with SCA, 17 controls	MP PC PS NO pro-ADM ET-1	To investigate the relationship between MP, PC, PS, NO, ET-1, and pro-ADM in paediatric patients with SCA.	-	Pro-ADM levels were elevated in acute chest syndrome versus steady state and controls.					During an acute chest crisis, ADM and ET-1 were elevated, suggesting a role for therapy inhibiting ET-1.
Robertson C.L. et al. (2001) [81]	CC, SC	1.5 mo–11 yr *n* = 31 21 patients with TBI, 10 controls without TBI or meningitis	CSF ADM	To investigate whether post- traumatic CSF ADM concentration was associated with relevant clinical variables (CBF).	Glasgow scale CBF measured using Xenon CT	ADM concentration was markedly increased in the CSF of infants and children after severe TBI versus controls. CBF increased one unit for every 0.893 unit increase in ADM concentration. Assessment of the relationship between ADM and outcomes did not reveal a significant association.					ADM may participate in the regulation of CBF after severe TBI.

The recorded results correspond to adrenomedullin. Sensitivity, specificity, and positive/negative predictive values are expressed as percentages. * Mean and standard deviation. AUC: area under curve; Se: sensitivity; Sp: specificity; PPV: positive predictive value; NPV: negative predictive value; CC: case control; SC: single centre; yr: years; ERG: electroretinogram; ADHD: attention deficit hyperactivity disorder; NO: nitric oxide; SCA: sickle cell anaemia; MP: circulating microparticles; PC: protein C; PS: free protein S; pro-ADM: proadrenomedullin; ET-1: endothelin-1; mo: months; TBI: traumatic brain injury; CSF: cerebrospinal fluid; CBF: cerebral blood flow; CT: computed tomography.

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
