# Peer review of "A Review of Adrenomedullin in Pediatric Patients: A Useful Biomarker"

_children, 2022, doi:10.3390/children9081181_

Round 1

Reviewer 1 Report

The present review article highlights the utility of adrenomedullin as a biomarker for several conditions in children. The subject is interesting and will be of interest to our readers. However, I have several issues that need to be addressed:

Introduction: What did you hypothesize before conducting this review? Please highlight this in 2-3 lines at the end of the introduction section.

Methods: The authors have mentioned that a "search of MEDLINE, PubMed, Embase, Web of Science, Scopus, and Cochrane was done." Was it a systematic search? If yes, please mention the PRISMA statement and give a PRISMA flow diagram. If not, it would be highly valuable if the authors perform a systematic literature search in the revised manuscript. A mere narrative review carries a high risk of bias.

-Why were studies on the adult population excluded?  How does age affect the ADM values?

Results: I would recommend the authors to perform a systematic review on this topic. The value of the findings is very limited. The risk of bias assessment with a grade of evidence also needs to be assessed.

Discussion: Adrenomedullin is a non-specific marker, elevated in a wide range of pathologies. Hence, the clinical utility of this biomarker would be limited. Therefore, the translational value of this review is limited. Nevertheless, I recommend the authors to perform a systematic review of the available literature.

Please mention the limitations of your review in one paragraph at the end of the discussion section.

Author Response

Dear Reviewer, thank you very much for your comments. Here are the answer to them.

Introduction: What did you hypothesize before conducting this review? Please highlight this in 2-3 lines at the end of the introduction section.

The hypothesis has been written at the end of the introduction section.

Methods: The authors have mentioned that a "search of MEDLINE, PubMed, Embase, Web of Science, Scopus, and Cochrane was done." Was it a systematic search? If yes, please mention the PRISMA statement and give a PRISMA flow diagram. If not, it would be highly valuable if the authors perform a systematic literature search in the revised manuscript. A mere narrative review carries a high risk of bias.

Yes, it is a systematic review. A PRISMA flow diagram was included as figure 1
and the PRISMA statement checklist as well as PRISMA for abstract checklist was
completed and uploaded. PRISMA protocol was also mention in the text (first paragraph
of methods).

-Why were studies on the adult population excluded?  How does age affect the ADM values?

Currently, there are many reviews on the usefulness of adrenomedullin in adults,
but there are no reviews exclusively focused on children. As paediatricians, we
believe that a review focused only on data from studies conducted on children is
essential. Children are not small adults and pediatric pathologies are different
from those of adults. That is why we focus our review exclusively on pediatric
studies.

Results: I would recommend the authors to perform a systematic review on this topic. The value of the findings is very limited. The risk of bias assessment with a grade of evidence also needs to be assessed.

The risk of bias of this systematic review was assessed by QUADAS-2, a graph
(figure 2) was included and it was explained at methods section (second
paragraph).

Discussion: Adrenomedullin is a non-specific marker, elevated in a wide range of pathologies. Hence, the clinical utility of this biomarker would be limited. Therefore, the translational value of this review is limited. Nevertheless, I recommend the authors to perform a systematic review of the available literature.

This is a systematic review. Nevertheless as assessed with QUADAS-2
recomendations bias may be present in many articles. This is commented
at results and discussion sections.

Please mention the limitations of your review in one paragraph at the end of the discussion section.

A paragraph explaining the limitations of the review was added at the end of the
discussion section. 

Reviewer 2 Report

Please find the comments and suggestions for the authors in the attached file.

Round 2

Reviewer 1 Report

I would like to congratulate the authors on their work. In the revised manuscript, the authors have addressed all my comments. The overall scientific quality of the review has improved significantly in the revised version. The study has merit and will be of interest to our readers.